# Modification of Rubberized Concrete: A Review

Changming Bu [1,2], Dongxu Zhu [1,2], Xinyu Lu [1,2], Lei Liu [1,2], Yi Sun [1,2,*], Linwen Yu [3], Tao Xiao [1] and Wentao Zhang [1]

1. School of Civil Engineering and Architecture, Chongqing University of Science & Technology, Chongqing 401331, China; buchangming@cqust.edu.cn (C.B.); 2020206029@cqust.edu.cn (D.Z.); 2020206077@cqust.edu.cn (X.L.); 2020206111@cqust.edu.cn (L.L.); 17318430251m@sina.cn (T.X.); kadiotao@gmail.com (W.Z.)
2. Chongqing Key Laboratory of Energy Engineering Mechanics & Disaster Prevention and Mitigation, Chongqing 401331, China
3. School of Materials Science and Engineering, Chongqing University, Chongqing 400044, China; linwen.yu@cqu.edu.cn
* Correspondence: sunyi@cqust.edu.cn; Tel.: +86-135-9416-9610

**Abstract:** One of the environmental problems causing concern in the world today is the black pollution caused by the accumulation of waste rubber resources. Relevant experimental studies have proved that rubber concrete can help solve the black pollution problem caused by waste rubber tires, but it is undeniable that rubber particles will reduce the mechanical properties of concrete. To this end, many studies on the modification of rubberized concrete have been carried out, and this paper summarizes these studies, considering compressive strength, durability performance and insulation performance. The results show that chemical pretreatments, such as sodium hydroxide, can significantly improve the adhesion between rubber particles and cement matrixes. Mineral powder admixtures, such as silica fume and fiber admixtures, e.g., PP fibers, can improve the compressive strength of rubber concrete.

**Keywords:** ceramsite concrete; thermal conductivity; durability; compressive strength; rubber

## 1. Introduction

The massive accumulation of waste rubber tires has caused a black pollution problem that threatens the ecological environment [1]. Research on rubber concrete can help solve the black pollution problem. The weak mechanical properties of rubber are responsible for the weak mechanical properties of rubber concrete which make it impossible for rubber concrete to be widely used. Properly solving the weakening of the mechanical properties of concrete due to rubber can expand the application range of rubber concrete and solve the problem of black pollution. The results show that the main reason for the weakening of the compressive strength of concrete after adding rubber is the bonding between rubber particles and a cement matrix, the bonding interface between them producing more micropores than that of ordinary concrete [2,3]. Under the action of load, the deformation of micropores leads to failure, which in turn leads to the failure of the overall structures in which they have been used. Therefore, many scholars began to study how to improve the interface transition zone between the two substances and enhance the bond strength between rubber particles and cement matrixes. The main reason why the rubber particles and the cement matrix are not tightly bonded is that the surface of the rubber particles is very smooth. Many scholars have proposed methods for modifying rubber particles, such as chemical soaking treatment and physical treatment. The chemical treatment involves roughening the rubber surface with strong corrosive or oxidative chemical solutions, such as NaOH and $KnMO_4$ [4–6]. Physical treatment is a special method for adsorbing small particles on rubber particles to improve the surface roughness of rubber.

Some scholars have not modified the rubber particles but have modified the rubber concrete by adding mineral admixtures [7–9], mainly high-strength retarders to prolong the hydration reaction of rubber concrete, as well as adding ultrafine fly ash to fill the micropores between the rubber particles and the cement matrix, etc. In addition to adding mineral admixtures, some scholars have proposed that rubber concrete can also be modified by adding fiber-reinforced rubber concrete. After extensive research and argumentation, both methods are considered feasible.

The durability of rubberized concrete [10,11] has also attracted much attention, especially its resistance to chloride ion penetration. Excellent resistance to chloride ion penetration helps protect structural elements such as steel bars in concrete, which can extend the life of building structures. At present, there are differences in the research on the durability change of rubberized concrete, and there is no unified conclusion in the research field. Some scholars [12,13] have shown that the excellent water resistance of rubber can enhance the chloride ion penetration resistance and penetration resistance of concrete, and this conclusion has been recognized by many scholars [11,14,15]. However, some scholars [16–18] believe that the micropores generated by the loose bonding between the rubber particles and the cement matrix will weaken the resistance of rubber concrete to the penetration of chloride ions and that the micropores will become channels for the penetration of chloride ions.

Another excellent property of rubberized concrete that has been recognized is its lower thermal conductivity [19–21] compared to conventional concrete. Rubberized concrete is an excellent thermal insulating concrete material. In the traditional prefabricated building field, in order to meet building insulation requirements, complex laminated wall panels are often produced, but rubber concrete can meet the building insulation requirements. The excellent thermal insulation properties of rubber particles are fully utilized.

Research on the compressive strength, durability and thermal conductivity of rubberized concrete is reviewed, and the progress of the research into rubber pretreatment and the addition of mineral admixtures and fibers is summarized. The research status of rubber concrete is expounded, and corresponding suggestions for follow-up research work are put forward.

## 2. Waste Rubber Particles

### 2.1. Type and Size of Rubber Particles

At present, there are as many as 25 kinds of rubber available on the market [1]. These kinds have different practical application fields. Automobile tires mainly use natural rubber, or natural rubber and carbon black composite materials, so the waste tire rubber involved in rubber concrete research is generally natural rubber or more complex synthetic rubber material. The utilization of waste tire rubber produced every year is shown in Table 1. Table 1 shows that 30% of the rubber is directly crushed and then landfilled. This treatment method not only wastes rubber resources, but also pollutes the environment.

**Table 1.** Statistics for world waste rubber resource utilization [1].

|  | Percentage |
|---|---|
| Recovery | 3–15% |
| Reutilization | 5–23% |
| Recovery (energy) | 25–60% |
| Stacking | 20–30% |

Forrest [3] pointed out that rubber can be recycled in many ways, but it cannot be widely used due to the high energy consumption caused by technology shortages at present. Forrest thinks it is possible to simply reprocess rubber for reuse, such as turning it into coarse particles that can be used as aggregate in concrete [2]. Roychand [22] found that rubber particles often play an unfavorable role in the failure process of rubberized concrete. He believes that this is due to the soft and elastic properties of rubber. The larger the size

of the rubber particles, the more space they occupy in the rubber concrete, and the more severe the failure response of the structure under load [23].

### 2.2. Properties of Rubber Particles

When conducting the test of rubber concrete, scholars choose different types and sizes of rubber particles, which are summarized in Table 2 for the convenience of readers.

**Table 2.** Rubber particle size statistics table.

| Reference | CR Size | Reference | CR Size | Reference | CR Size |
|---|---|---|---|---|---|
| Steyn et al. [24] | <4.75 mm (fine) | Taak et al. [25] | 10–20 mm (coarse) | Dehdezi et al. [26] | 2–4 mm (fine) |
| Tiwari et al. [27] | <4.75 mm (fine) | Mhaya et al. [9] | 1–4 mm (fine), 5–8 mm (coarse) | Abdelmonem et al. [28] | 0–4 mm (fine) |
| Ramdani et al. [29] | 0.2–4 mm (fine) | Karunarathna et al. [23] | 2–4 mm (fine), 15 mm(coarse) | Othman et al. [30] | 180 μm (fine) |
| Ossola. [31] | 420–840 μm (fine) | Rahim et al. [32] | 2–4 mm (fine) | Mousavimehr et al. [33] | 0–4.75 mm (fine) |
| Barrera et al. [34] | 0.85–2.8 mm (fine) | Habib et al. [35] | 0.075–10 mm (fine) | Liu et al. [36] | <0.42 mm (fine) |
| Su et al. [37] | 0.3 mm, 0.5 mm, 3 mm (fine) | Chaikaew et al. [38] | 3.36 mm (fine) | Eisa et al. [39] | 2–3 mm (fine) |
| Zhu et al. [40] | 1–2 mm (fine) | Aslani et al. [41] | 2–5 mm (fine) | Shahjalal et al. [42] | 4.75–19 mm (fine\coarse) |
| Li et al. [43] | 0–4 mm (fine) | Wang et al. [44] | 1.19–19 mm (fine\coarse) | Gerges et al. [45] | 710 μm (fine) |
| Aureliano et al. [46] | 0–1.18 mm (fine) | He et al. [47] | <0.088 mm (fine) | Liu et al. [48] | 0.178, 1.11, 2 mm (fine) |

The smoothness of rubber prevents it from bonding effectively to cementitious substrates. Delilah [49] put rubber particles into a 0.1 m sodium carbonate buffer containing 3% glutaraldehyde and 2% formaldehyde for special treatment, and the surface of the rubber particles after the removal of impurities was smooth, as shown in Figure 1. In fact, if no special treatment is performed, but only after pulverization, the surface of the rubber particles will appear to be not smooth [50], as shown in Figure 2. The impurities remaining on the surface of the rubber particles after crushing will fall off when combined with the cement matrix, and there is no tight bond between the rubber particles and the impurities. Deliah's research revealed the true appearance of rubber particles, and Segre [51] also observed the same phenomenon.

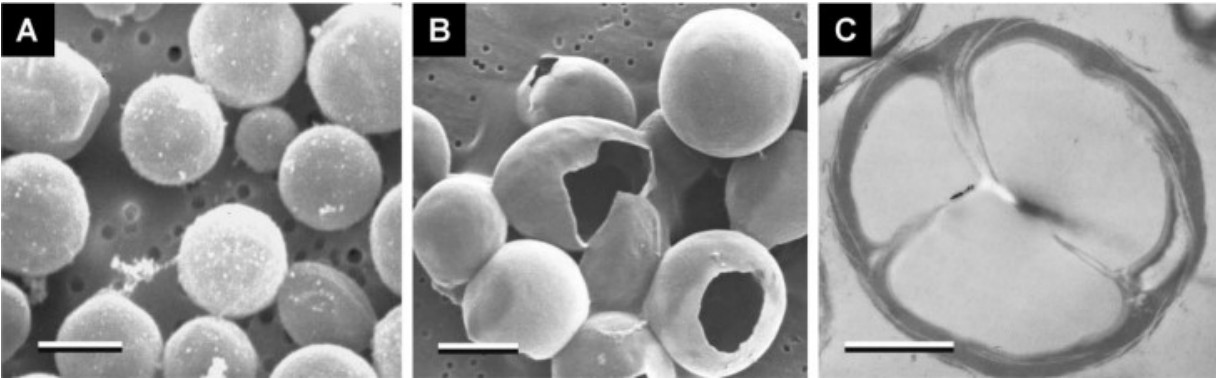

**Figure 1.** Microscopic image of elastic rubber particles. (**A**) Conventional SEM. (**B**) Field emission SEM at room temperature. (**C**) Transmission electron micrograph (TEM) of single rubber particles [49].

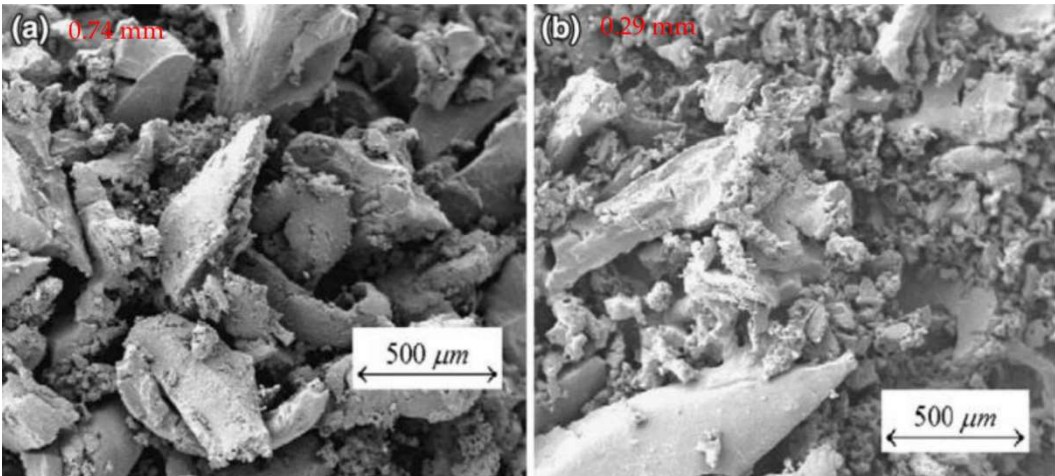

**Figure 2.** SEM of crumb rubber [50]. (**a**) 0.74 mm (**b**) 0.29 mm.

Delilah [49] pointed out that these impurities include acidic substances and carboxyl groups, which are derived from various admixtures added during the preparation of rubber tires. Rubber tires go through an intensive refining process during the production process, in which oils, additives, accelerators, carbon black and other substances are added. The composition of rubber tires is shown in Table 3, and the substances added during this process cannot be removed by simple mechanical crushing [22], as shown in Figure 3. Xiao et al. [52] found that low temperature plasma treatment roughened the surface of rubber particles and greatly reduced the water contact angle of rubber. The rough surface of the rubber was able to bond with the cement matrix better.

Rubber is a highly elastic material that can recover when subjected to external forces. The high elastic properties of rubber can make up for the high brittleness of concrete, while the low elastic modulus of concrete has always been the focus of research. The work of Karunarathna [23] shows that the elastic modulus of concrete is obviously reduced after the incorporation of rubber, and rubber can delay the development of structural cracks. Meanwhile, rubber particles can also act as bridges between cracks when concrete is damaged by force, as shown in Figure 4.

**Table 3.** Basic composition of reclaimed rubber particles [22].

| Material | Main Ingredients | Composition |
| --- | --- | --- |
| Rubber | Natural rubber, synthetic rubber | 51% |
| Reinforcing agent | Carbon black, silica | 25% |
| Softener | Petroleum process oil, petroleum synthetic resin, etc. | 19.5% |
| Vulcanizing agent | Sulphur, organic vulcanizers | 1.0% |
| Vulcanizing accelerator | Thiazole accelerators, sulfenic amide accelerator | 1.5% |
| Vulcanizing accelerator aid | Zinc oxide, stearic acid | 0.5% |
| Antioxidant | Amine antioxidants, phenol antioxidants, wax | 1.5% |
| Filler | Calcium carbonate, clay | |

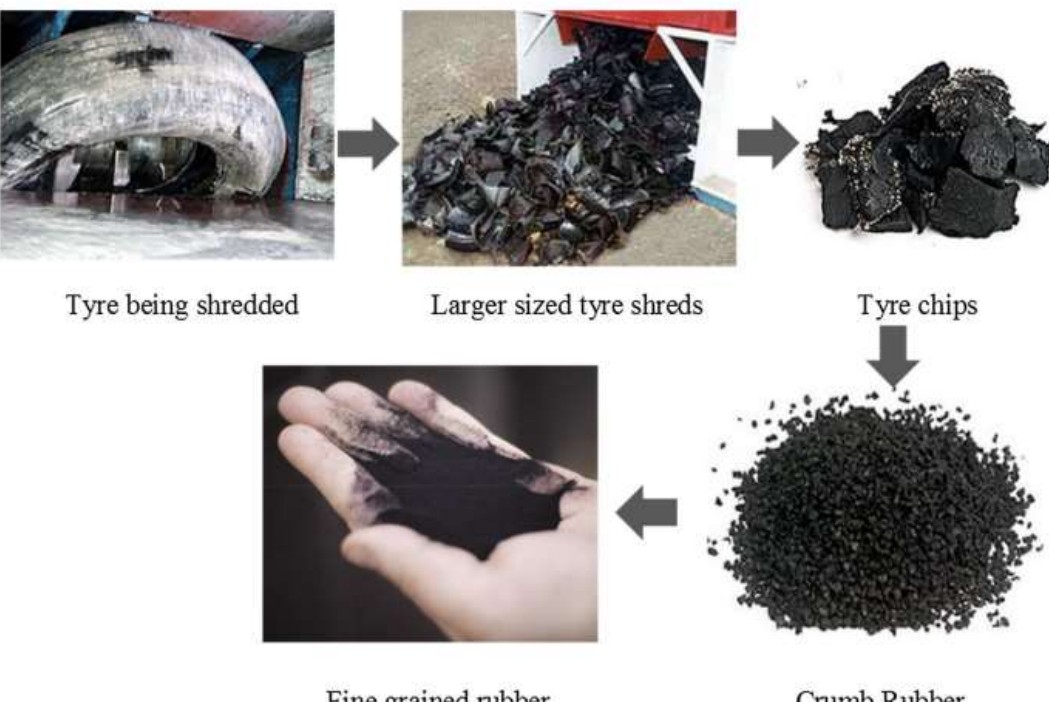

**Figure 3.** Different stages of tire shredding [22].

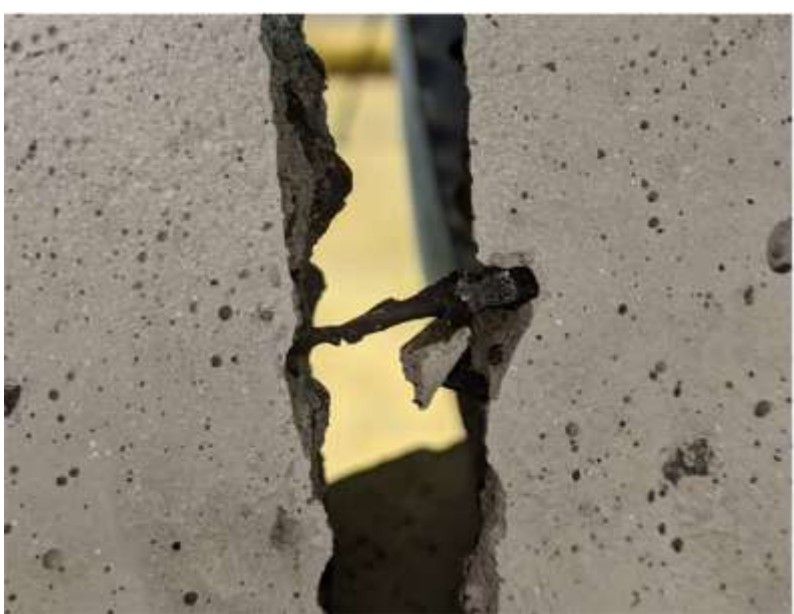

**Figure 4.** Crack bridging by tire shreds [23].

Rubber is a kind of porous material. Karunarathna confirmed the high gas content of rubber through research and photographed overflow bubbles on rubber surfaces when soaked in water, as shown in Figure 5. The porous structure of rubber allows it to contain more air, resulting in weak adhesion between rubber and cement matrix when mixed with concrete. Roychand [22] expressed that the problem of high gas content in rubber can be effectively improved by soaking the rubber in advance. Roychand believes that the porous structure in rubber is derived from the linear structure of the internal material composition, which guarantees the high elastic properties of the rubber.

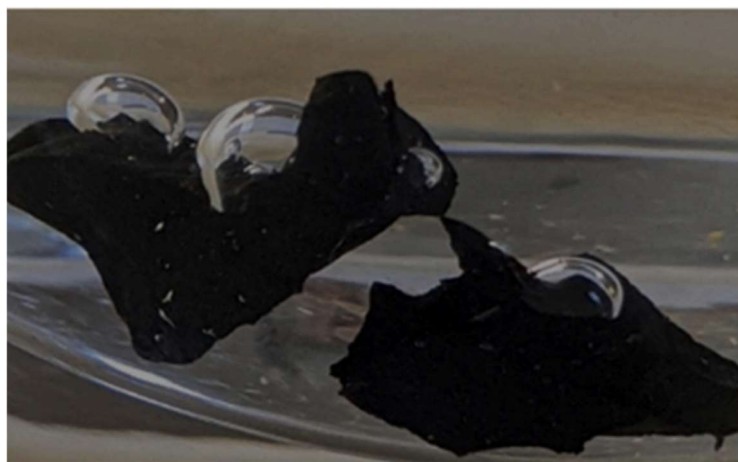

**Figure 5.** Trapped air bubbles on tire shreds submerged in water [23].

Through Fourier transform infrared and X-ray fluorescence analysis, Jusli et al. [53] found that the main chemical components of broken tire rubber particles were carbon, zinc, silicon, magnesium and calcium. The main chemical components are shown in Table 4. According to Table 4, except for SBR, carbon black accounts for the vast majority. In addition, there are admixtures such as oil that cause crushed tire rubber to act differently to rubber alone. This is also a factor to consider when recycling scrap tire rubber. Delilah et al. [49] purified natural rubber using lactic acid bacteria and latex and concluded that natural rubber is composed of a rubber core and a protein coating by low-temperature plasma monorail scanning and protein chemical analysis, as shown in Figure 6.

**Table 4.** Rubber chemical composition [53].

| Chemical Composition | Percentage (%) |
| --- | --- |
| SBR | 48.0 |
| Carbon black | 47.0 |
| Extender oil | 1.9 |
| Zinc oxide | 1.1 |
| Stearic acid | 0.5 |
| Sulfur | 0.8 |
| Accelerator | 0.7 |

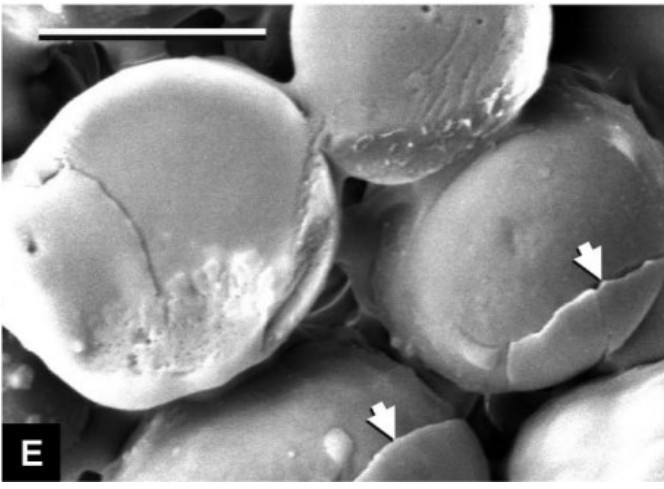

**Figure 6.** Rubber structure diagram: low-temperature field-emission SEM (at 4 kV) of frozen-hydrated fractured particles; magnification bar = 2 mm [49].

## 3. Rubber Pretreatment

Navarro [50] showed that when rubber tires are produced they are mixed with admixtures, including carbon black, and these impurities will have adverse effects on the adhesion of rubber particles to cement matrixes, which is an important reason why rubber particles from waste tires need additional treatment.

Jokar et al. [4] used NaOH solution to soak rubber particles to explore the performance difference between treated rubberized concrete and ordinary rubberized concrete. The results show that the rubber particles pretreated with NaOH have better bonding relationships with cement matrixes and higher compressive strength, as shown in Figure 7. The NaOH solution not only helps to remove impurities from the rubber particles, such as carbon black, which cannot be removed mechanically when the tire is broken into rubber particles, but also roughens the rubber surface, thereby optimizing the bond between the rubber and the cement matrix.

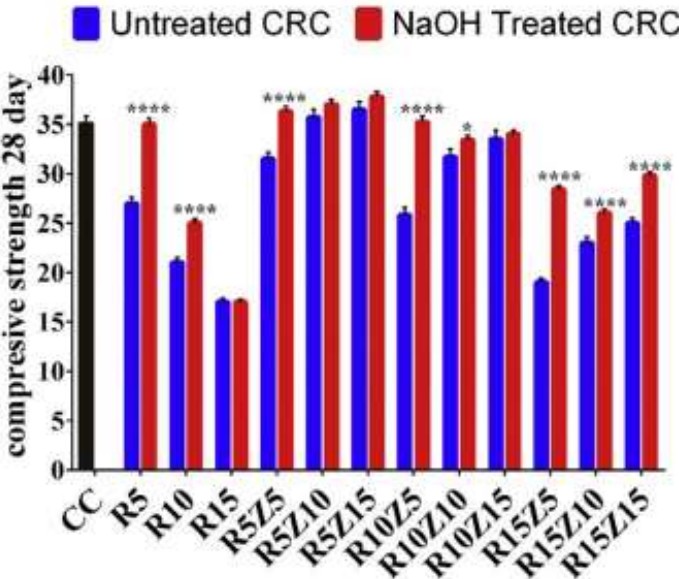

**Figure 7.** Comparison chart of the compressive strength of pretreated and untreated rubberized concrete at 28 days (MPa) (R represents the replacement rate of rubber; Z represents the replacement rate of zeolite) (* $p < 0.05$, ** $p < 0.001$, *** $p < 0.0001$, **** $p < 0.00001$) [4].

Qin et al. [19] focused on another kind of waste silicone rubber material derived from insulators, which is different from the styrene butadiene rubber of waste automobile tires. The composition of silicone rubber is shown in Figure 8. Qin used hydrogen peroxide solution and potassium hydroxide solution, respectively, to conduct modification tests on silicone rubber. The compressive strength test shows that the rubber concrete with special treatment had a better performance, as shown in Figure 9. By testing water contact angle, it was found that the silicone rubber particles with special treatment had better water absorption performance than untreated silicone rubber.

Kumar and Dev [5] pretreated rubber particles with sulfuric acid and analyzed chemical composition changes in rubber particles before and after treatment using EDX, as shown in Table 5. On this basis, the compressive strength performance of the pretreated rubber concrete was compared with that of the untreated rubber concrete. The results showed that the treated rubber particles could produce better bonds with the cement matrix and yield higher compressive strength.

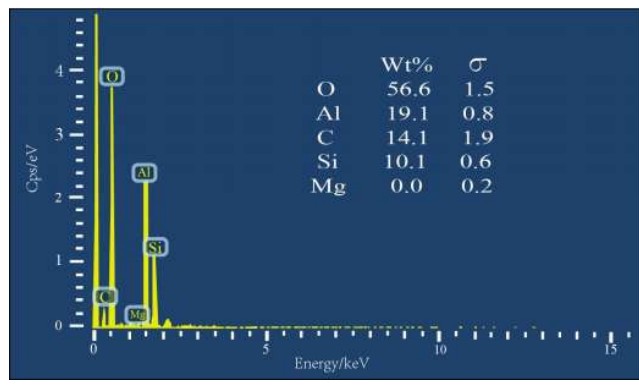

**Figure 8.** Composition table of silicone rubber (EDS) [19].

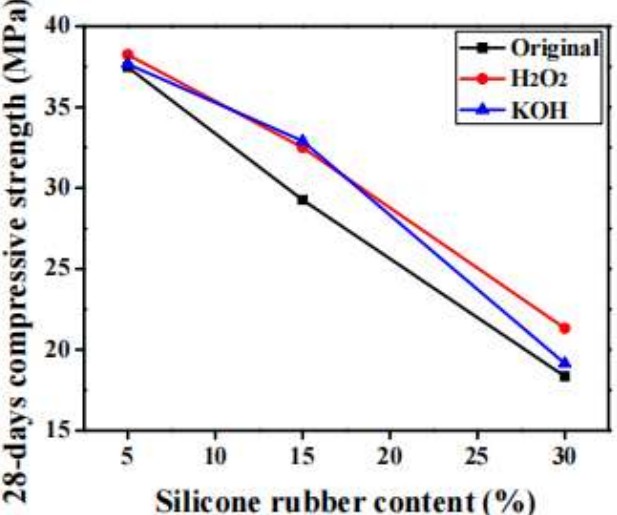

**Figure 9.** Compressive strength of silicone rubber mortar at 28 days [19].

**Table 5.** Chemical elements in rubber crumb before and after surface treatment in EDX analysis [5].

| Element | Before Treatment of Rubber | | After Treatment of Rubber | |
|---|---|---|---|---|
| | Weight% | Atomic% | Weight% | Atomic% |
| C K | 62.80 | 71.52 | 70.25 | 78.10 |
| O K | 29.92 | 25.58 | 22.71 | 18.95 |
| Al K | 0.80 | 0.40 | - | - |
| Si K | 2.39 | 1.16 | 3.20 | 1.52 |
| S K | 0.64 | 0.27 | 1.74 | 0.72 |
| Ca K | 2.56 | 0.87 | 2.10 | 0.70 |
| Zn K | 0.88 | 0.18 | - | - |
| Total | 100 | | 100 | |

Zhang et al. [54] used acrylic acid and polyethylene glycol to modify the surface of rubber particles. ACA and PEG can polymerize hydrophilic functional groups and improve the hydrophobic properties of rubber particles. The improvement was expressed in terms of water contact angle, as shown in Figure 10. Through microscopic research, it was found that the modified rubber particles could more effectively combine with the cement matrix, which showed that the modified rubber concrete had stronger compressive strength. Najim [55] also came to the same conclusion.

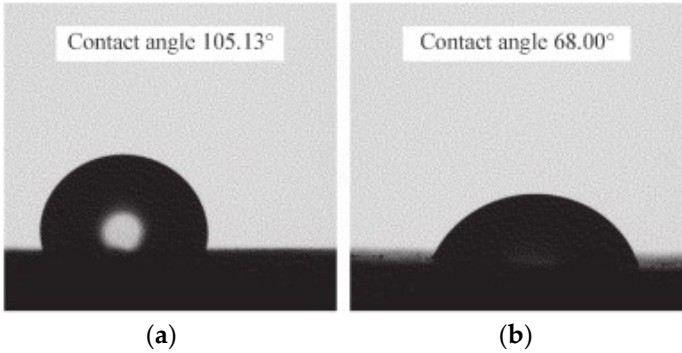

**Figure 10.** Effect of modification on distilled water wetting contact angle of rubber [54]. (**a**) Before modification. (**b**) After modification.

　　Youssf et al. [6] studied modification effects using NaOH, $H_2O_2$, $H_2SO_4$, $CaCl_2$, $KMnO_4$, $NaHsO_3$ and silane coupling agent on rubber concrete. The compressive strength of the pretreated rubber concrete is shown in Figure 11. It can be observed that the pretreatment with NaOH and $CaCl_2$ gave the best results; in contrast, the acid pretreatment showed little improvement of the compressive strength of the rubberized concrete. In particular, the compressive strength of rubber concrete pretreated with potassium permanganate and sodium hydrogen sulfate solution was weaker than that of ordinary rubber concrete without any treatment. The author believes that this is because the alkaline hydration reaction environment in concrete will be affected by acidic substances.

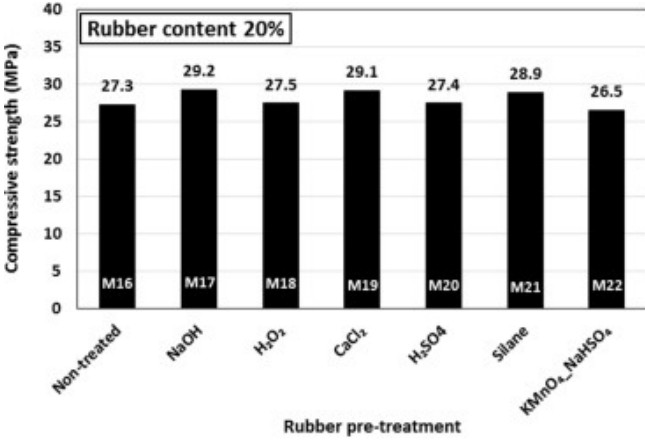

**Figure 11.** Effects of rubber pretreatment on rubberized concrete strength [6].

　　Zhu [56] carried out research on improving the transition zone of the rubber concrete interface. Zhu pretreated rubber with a silane modifier and explored the bond situation between the specially treated rubber and the cement matrix. The test results showed that the rubber treated with the silane modifier could better bond with the cement matrix, which was due to the wet performance and bonding performance of the silane modifier itself. Chen et al. [57] used sodium hydroxide and ethyl orthosilicate to modify rubber concrete; the results showed that the modified rubber concrete had better compressive strength performance. The author studied the microstructure of the rubber concrete using SEM and found that a large number of hydrophilic functional groups were passively accepted in the process of rubber modification, which was an important factor in the enhancement of the bonding between the rubber and the cement matrix. Secondly, the hydrolysis and condensation reactions of these functional groups strengthened the bonds between the rubber and the cement matrix, which led to the improvement of the compressive strength of the rubber concrete. The principle of functional group action is shown in Figure 12.

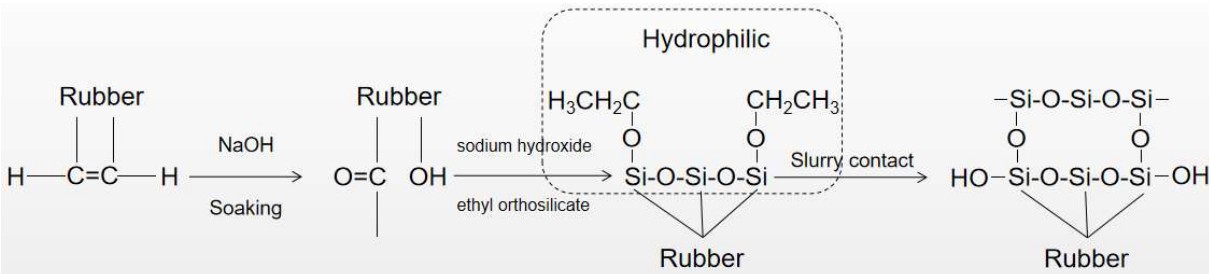

**Figure 12.** Schematic diagram of chemical modification of rubber surface and contact with cement paste.

Pham et al. [10] Pretreated rubber with a copolymer composite coating and explored the performance of pretreated rubberized concrete in resisting freeze–thaw cycles. The test results showed that the anti-freeze–thaw cycle performance of the coated rubberized concrete was weaker than that of the original rubberized concrete, but the mass loss of both after the same freeze–thaw cycle was much higher than that of ordinary concrete. This shows that rubberized concrete has excellent resistance to freeze–thaw cycles and has application potential in cold regions. The residual compressive strength of the rubberized concrete after freeze–thaw cycles was higher than that of the untreated rubberized concrete. This shows the excellent performance of rubber concrete in terms of concrete durability. The test results are shown in Figures 13 and 14. The author believes that the results were determined by the characteristics of the rubber itself. The stable expansion and shrinkage resistance of the rubber particles can help a concrete specimen to release the expansion and contraction pressure caused by the freezing and thawing of water molecules during the freeze–thaw cycle. The pretreatment process can help the rubberized concrete retain more residual strength.

Kashani et al. [11] studied five pretreatment methods for rubber particles, namely, sodium hydroxide, potassium permanganate, sulfuric acid, silica fume coating and cement coating. The results show that sodium hydroxide, potassium permanganate and sulfuric acid solutions can improve the water contact angle of rubber particles, effectively improve the bonding between rubber and cement matrix, and reduce the internal porosity of rubber concrete, which helps rubber concrete with chlorine ion penetration. Zhong et al. [58] used styrene–acrylic emulsion to impregnate rubber particles and the performance of the specially treated rubberized concrete during freeze–thaw cycles was studied. The test results showed that the rubber concrete after special treatment had a better compressive strength residual than ordinary rubber concrete after 300 freeze–thaw cycles. Chou et al. [59] used organic sulfur to optimize the bonding interface of rubber concrete. The author thinks that the treatment of organic sulfur can help the rubber particles to bond with the cement matrix better and optimize the microstructure of rubber concrete. Chaturvedy et al. [12] reviewed the common points in the research on modified rubber concrete and considered that improving the bonding interface of rubber and cement matrix is an important factor in optimizing the durability performance of rubber concrete, the modification mechanism of chemical pretreatment being the most important factor.

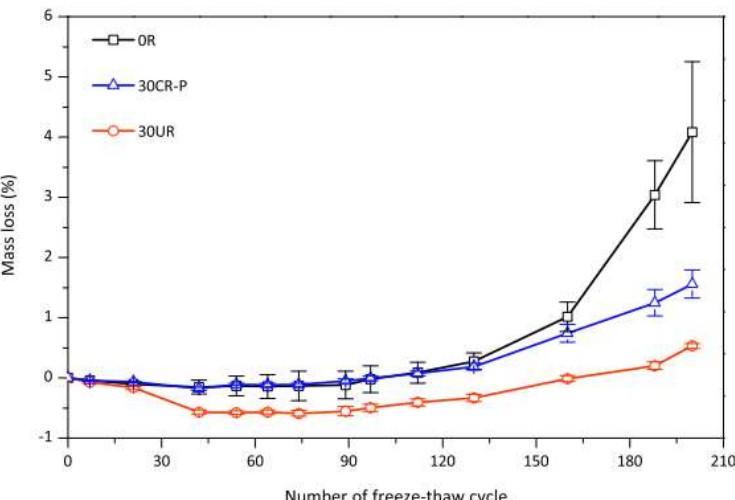

**Figure 13.** Comparison of mass loss of common mortar and rubber mortar after freeze−thaw cycle [10].

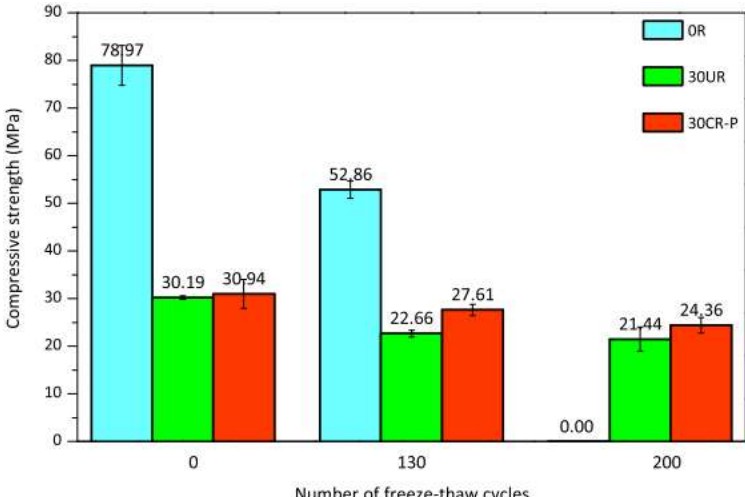

**Figure 14.** Variation diagram of compressive strength of samples with the increase in freeze–thaw cycles [10].

The use of chemical products to modify rubber particles has potential to improve the compressive strength and durability of rubber concrete. The improvement of the adhesion between rubber particles and cement matrix is the key to improving mechanical properties and durability. The surface modification of rubber particles by solutions such as sodium hydroxide and the filling of pores by materials such as styrene–acrylic emulsions can effectively optimize the bonding between the particles and the cement matrix, effectively reduce the pores inside the rubber concrete and improve compressive strength and durability. In fact, even though the strength of pretreated rubber concrete cannot be compared with that of the control without rubber, the strength of pretreated rubber concrete is significantly improved compared with that of untreated rubber concrete. The degree of improvement is affected by many factors, such as treatment method, treatment substance and rubber particle size. The different methods of rubber pretreatment selected by different scholars and the effects achieved after treatment are summarized in Table 6.

**Table 6.** Summary table of rubber pretreatment.

| References | Rubber Particle Size | Best Rubber Content | Treatment Methods | Compressive Strength | Durability |
|---|---|---|---|---|---|
| Navarro [50] | Average particle diameter is 0.63 mm | 9 wt% | Mix in an open low shear intermittent mixer for 1.5 h at 180 centigrade and 1200 rpm | – | – |
| Jokar [4] | 1–6 mm | 5 wt% rubber and 15% zeolite | 1. Soak in 1 M NaOH aqueous solution for 24 h, then rinse the rubber particles with clean water and dry them at room temperature 2. Add zeolite | The compressive strength of all samples of rubber concrete modified by NaOH solution at 7 and 28 days was higher than that of the untreated control group | – |
| Qin [19] | 5, 10, 20 and 50 mesh | 5 wt% treated with KOH or $H_2O_2$ | 1. Soak in $H_2O_2$ for 24 H at room temperature 2. Soak in KOH for 24 H and then wash with water several times | With the increase in rubber content, the compressive strength of mortar decreased slightly. The pretreatment method has an enhancing effect on mechanical strength of mortar | – |
| Kumar [5] | 0.6–2.36 mm | 15 wt% | The rubber particles were first soaked in 15 wt% sulfuric acid solution for 2 h, then washed with water 3–4 times and finally dried by natural air for 24 h | The rubber particles pretreated with sulfuric acid will not cause serious strength loss even when the content reaches 15 wt% | – |
| Zhang [54] | 5 and 40 mesh | For impact resistance, 5 wt% is best | The modifier prepared by 17.2 wt% aca, 13.8 wt% peg and 69.0 wt% ae is sprayed on the surface of rubber particles, then stirred for 20 min, then placed in an oven and heated at 40 °C for 30 min, then heated to 110 °C for 45 min and finally naturally cooled to room temperature | Rubber particle modification can improve compressive strength | Modified rubber particles can reduce the water–cement ratio of concrete mixtures from 0.4 to 0.38 |

**Table 6.** *Cont.*

| References | Rubber Particle Size | Best Rubber Content | Treatment Methods | Compressive Strength | Durability |
|---|---|---|---|---|---|
| Najim [55] | 2–6 mm | 12% with mortar precoating | 1. Water washing 2. Cement paste precoating 3. Mortar precoating 4. NaOH pretreatment | Precoating rubber particles with mortar can significantly improve the interfacial bonding between interphase rubber and cement mortar, and the compressive strength can be increased by 37% | Mortar pretreatment gives rubber concrete a better hole distribution, although the size of holes is not very different from that of other pretreatment methods |
| Youssf [6] | 2.36 and 4.75 mm | 15 wt% | 1. Water washing 2. Water soaking-A 3. Water soaking-O 4. NaOH 5. $H_2O_2$ 6. $CaCl_2$ 7. $H_2SO_4$ 8. Silane 9. $KMnO_4\_NaHSO_4$ | After adding rubber, the compressive strength of concrete decreases obviously. The compressive strength of rubber concrete was improved by 2–10% by various pretreatment methods | Various pretreatment methods can help improve the hydrophilicity of rubber, which is manifested by a decrease in carbon content and an increase in oxygen content |
| Zhu [56] | 1–3 mm, 3–6 mm, 20 mesh | 5 wt% with the size is 3–6 mm | The surface properties of rubber particles were changed by immersing rubber in silane coupling agent | The compressive strength of rubber concrete decreases with the increase in rubber content. The larger the rubber particle size, the less the strength decreases; rubber particles with particle size of 1–3 mm have the best modification effect | After adding rubber, the total porosity increased; the larger the particle size of rubber, the smaller the median pore size; compared with the control group, the specific pore volume of the modified rubber concrete decreased significantly |
| Chen [57] | 300–700 μm | – | It was first immersed in sodium hydroxide solution and then immersed in ethyl orthosilicate ethanol solution prepared with ethyl orthosilicate (TEOS) and absolute ethanol solution for modification | The compressive strength of 10% rubber mortar modified with 5 wt% TEOS at 28 days was 28.58 MPa, which was 26.63% higher than that of the original rubber, greatly improving the compressive strength of the mortar | — |
| Pham [10] | 0.6–1.3 mm | 30 wt% rubber precoat with styrene butadiene-type copolymer | First, RA needs to be precoated with styrene butadiene copolymer (2% of RA mass). The treated RA was kept in a room at a temperature of 20 °C and a relative humidity of 50% for 1 h | – | The ability of resisting freeze–thaw cycles of rubber cement mortar was better than that of the control group, which shows that the residual strength was greater and the mass loss smaller. The combination with special treatment performed better |

**Table 6.** *Cont.*

| References | Rubber Particle Size | Best Rubber Content | Treatment Methods | Compressive Strength | Durability |
|---|---|---|---|---|---|
| Kashani [11] | 2.36–4.75 mm | 10 wt% with cement coating | 1. Two coatings (cement coating, silica fume coating) 2. Soaking (KMnO$_4$, NaOH, H$_2$SO$_4$) | Compared with untreated samples, the compressive strength of rubber concrete pretreated with different coating methods increased by 42% and 49% | Chemical pretreatment changed the hydrophobic properties of rubber particles, strengthened the bonds between rubber particles and cement matrix, and reduced the porosity of the specimen |
| Zhong [58] | 80 mesh | 10 wt% | 1. Washing 2. Soaking in NaOH solution 3. Modification by styrene–acrylic emulsion | When the rubber content was less than 20%, the modification effect of NaOH was the best; when the content of NaOH was less than 15%, the modification effect of NaOH was the best, followed by styrene–acrylic emulsion, while there was no significant difference between the water washing group and the control group | On the basis of the apparent analysis, it is believed that the addition of rubber particles helps to improve the ability of magnesium oxide cement to resist freeze–thaw cycles. |
| Chou [59] | 30–50 mesh | – | Soak the rubber particles in carbon disulfide and put them in glassware to stand at room temperature. After the carbon disulfide evaporates, clean the rubber with acetone and dissolved water | The pretreatment of carbon disulfide can help the rubber particles change the hydrophobic characteristics, strengthen the hydration process and enhance the compressive strength of rubber concrete | The pretreatment of carbon disulfide can help to improve the friction between the aggregate and the cement matrix in the concrete, strengthen the molecular forces between C-S-H, strengthen the hydration reaction and reduce porosity |

## 4. Incorporating External Compounds

Xue and Shinozuka [7] found that adding silica fume (SF) to rubberized concrete would yield higher compressive strength. The form of silica fume and rubber is shown in Figure 15, and the compressive strength comparison is shown in Figure 16. At the same time, the seismic disturbance resistance of rubberized concrete was improved compared with that of traditional concrete. The authors believe that this is because the silica fume fills the pores caused by the inconsistency of the rubber and cement matrix during the early mixing and curing process.

Li et al. [8] studied the modification mechanism of different materials in rubberized concrete and finally selected the composite admixture composed of silica fume, silane and polymer modifiers to modify rubberized concrete. The modification effect of the rubberized concrete after the incorporation of a small amount of composite admixture was significant.

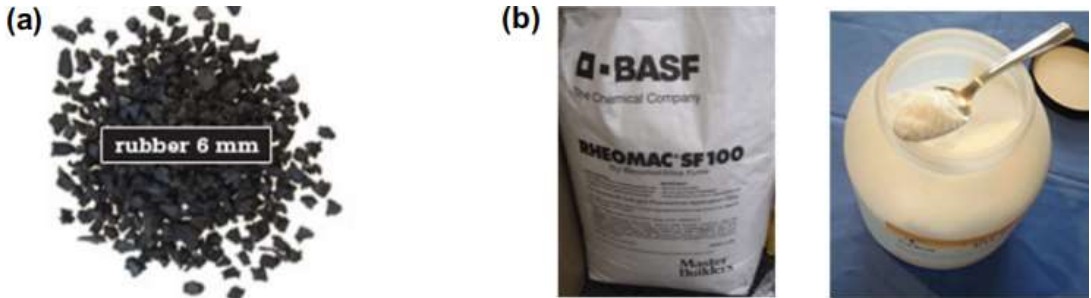

**Figure 15.** Material samples selected for test samples. (**a**) Rubber crumbs. (**b**) Silicon powder [7].

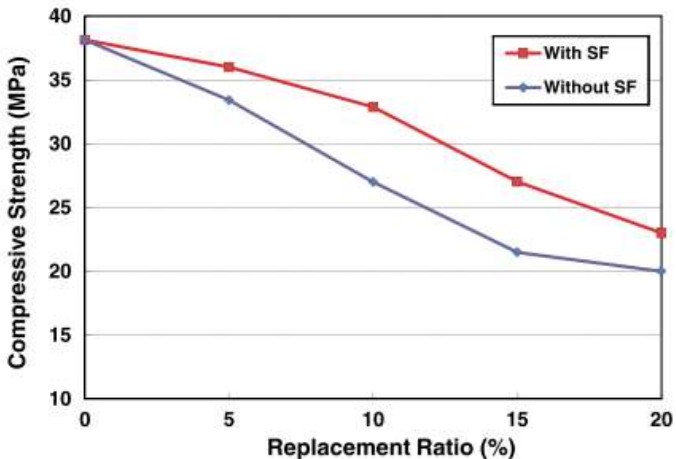

**Figure 16.** Comparison of compressive strength of test samples [7].

Hamid et al. [13] studied the mechanical properties of rubber concrete mixed with seawater mixture and designed 13 different seawater rubber concrete test groups. The test results are shown in Figure 17. Regarding the compressive strength of concrete mixed with coarse and fine rubber particles after mixing with seawater, a decreasing trend was first observed, followed by an increasing trend. Mhaya et al. [9] studied the effect of the addition of granulated blast furnace slag particles (GBFS) and rubber on the compressive strength of concrete. The test results showed that the rubber concrete mixed with GBFS showed better compressive strength. Through microscopic examination, it was found that the bonding between rubber and cement matrix was optimized after the incorporation of GBFS. After consideration, it is believed that the activation characteristics of GBFS not only act on the cement, but also cause a certain amount of corrosion to the surface of the rubber particles. The rough rubber particles can bond more closely with the cement matrix. The chemical composition and physical properties of GBFS and rubber are shown in Table 7.

Grinys et al. [60] modified rubberized concrete by adding glass powder and rubber latex. The results showed that although the strength of rubberized concrete mixed with glass powder did not change much in the early stage, the strength at 28 d was 11–13% higher than the rubber concrete without glass powder. On the basis of the observation of the microstructure, it is believed that the existence of glass powder activates the cement, so that the hydration reaction can continue, and a higher compressive strength appears in the later stage. The pozzolanic effect of the glass powder is also an important reason for the later strength improvement. Figure 18 shows the existing form of waste porous glass in concrete. Jokar et al. [4] modified rubber concrete by adding zeolite. The test results showed that the rubber concrete mixed with zeolite had a higher compressive strength than ordinary rubber concrete. The authors believe that this is because the pozzolanic effect and activation characteristics of the fine zeolite particles make the rubber particles and cement have better adhesion, which is manifested as improved compressive strength.

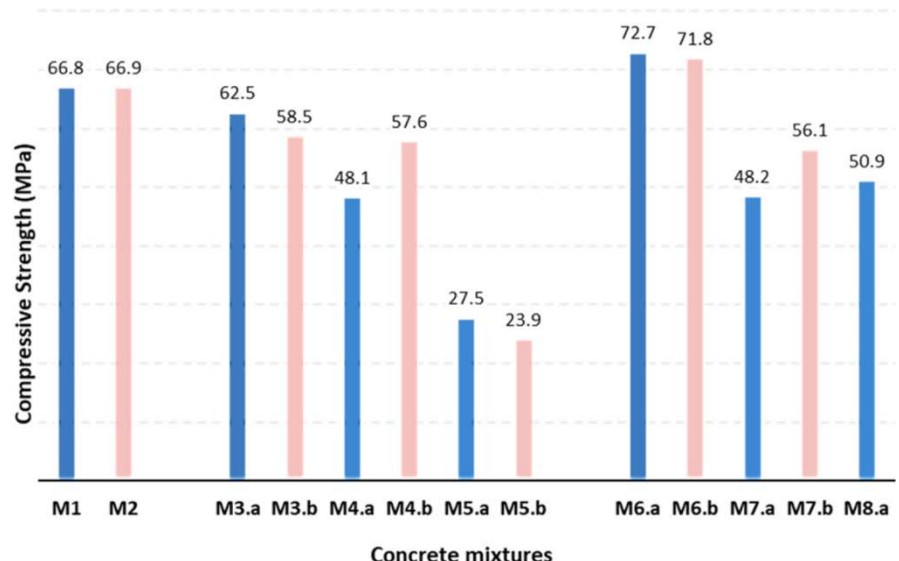

**Figure 17.** Comparison of the compressive strengths of blast furnace slag seawater concrete at 28 days curing age (M1 is the control group; M2 is the 100% seawater combination; M3 to M5 are fine aggregate rubber replacement groups; M6 to m8 are coarse aggregate rubber replacement groups) [13].

**Table 7.** Chemical and physical properties of blast furnace slag and rubber (XRF) [9].

| Chemical Compositions | GBFS (%) | Chemical Compositions | RC (%) | Physical Properties | RC |
|---|---|---|---|---|---|
| Silica Oxide, $SiO_2$ | 30.80 | Acetone extract | 10 | Size of fine rubber, mm | 1–4 |
| Aluminium Oxide, $Al_2O_3$ | 10.9 | Ash content | 24 | Size of coarse rubber, mm | 5–8 |
| Iron Oxide, $Fe_2O_3$ | 0.64 | Carbon black | 14 | Heat loss, $kgf/cm^2$ | <1 |
| Calcium Oxide, CaO | 51.80 | Rubber Hydrocarbon (RHC) | 52 | Metal content, % | <0.5 |
| Magnesium Oxide, MgO | 4.57 | | | Fiber content, % | <1 |
| Potassium Oxide, $K_2O$ | 0.36 | | | | |
| Loss on Ignition, LOI | 0.22 | | | | |

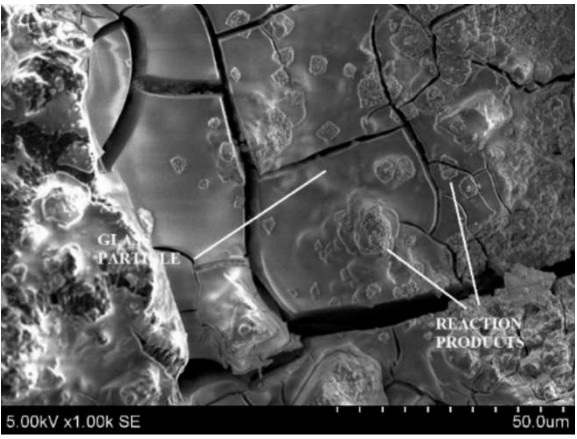

**Figure 18.** SEM images of glass waste particles [60].

Kang et al. [61] studied the change in compressive strength of rubberized concrete after adding silica fume. The test results showed that the addition of microsilica fume filled the tiny pores inside the rubberized concrete, and the compressive strength of the rubberized concrete was improved, which is consistent with the research results of many scholars [11,16,62–64]. The effects of different silica fume dosages on the compressive strength of rubberized concrete are shown in Figure 19. Li [65] used carbon nanotubes to

modify rubber cement mortar to explore the effect of carbon nanotubes on the compressive strength of rubber cement mortar. The results showed that the effect of carbon nanotubes on the compressive strength of rubber cement mortar was positive because the bridging effect of carbon nanotubes and the nanoforest effect can help to connect the rubber and cement matrix. At the same time, carbon nanotubes can also help strengthen the hydration reaction of cement, which is also an important reason for the improvement in compressive strength. Bashar et al. [66] incorporated nanosilica materials into rubber concrete, and the test results show that the rubber concrete incorporating nanosilica had a higher compressive strength. This because nanosilica filled the pores inside the rubberized concrete specimens, the reason being similar to the mechanism of action indicated by the study of Li [65].

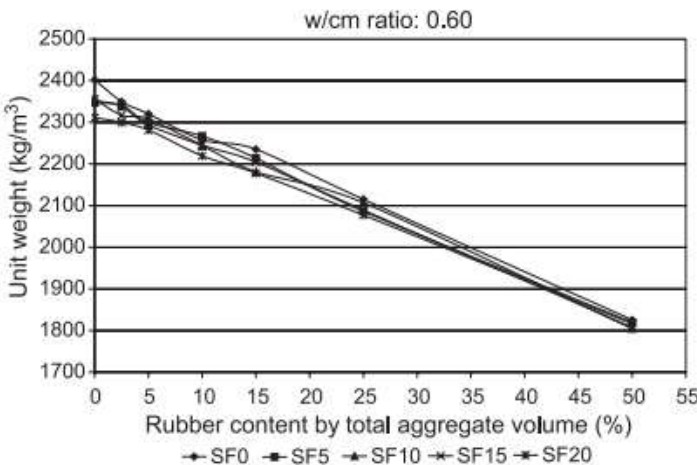

**Figure 19.** Influence of rubber and silica fume content on unit weight of concrete [62].

Zhang [67] studied the role of metakaolin (MK) in rubberized concrete, and the results showed that although the pozzolanic effect of MK helped to significantly improve the mechanical properties of rubberized concrete, the effect was not linear, first increasing first and then decreasing. Zhang thought that this was because MK is rich in activated alumina, which promotes secondary hydration, and the resulting calcium hydroxide can help build a denser internal system and resist greater external loads; however, a large amount of MK does not guarantee that the secondary hydration reaction will be enhanced, so there is a theoretical maximum value. The author suggests that the dosage should between 13–16%.

The study by Hamid et al. [13] also showed that the rubber concrete mixed with seawater showed a slight positive result in terms of water resistance. Combined with the compressive strength analysis, it can be considered that the presence of seawater helps the roughening of the rubber, which improves the strength of the bonding between rubber particles and cement matrix. Grinys [60] also investigated the effect of glass powder on the durability of rubber concrete and obtained the residual compressive strengths of different test groups through 200 freeze–thaw cycles. The results showed that the freeze–thaw resistance of concrete was greatly improved after incorporating rubber, but the ability of rubber concrete to resist freeze–thaw cycles was inhibited when glass powder was added. This was because the rubber selected for the preparation of tires is almost non-polar, and when combined with the cement matrix, many tiny pores will be generated. The air stored in these pores will help the rubber concrete release the expansion and contraction pressure generated in the freezing and thawing environment, but the filling effect of glass frit and the pozzolanic effect will fill these pores. Although this will help the compressive strength of rubberized concrete, it will reduce its freeze–thaw resistance. Blast furnace slag (GGBS), metakaolin (MK) and fly ash (FA) were added to rubberized concrete by Siad et al. [17] to explore the effects of these three mineral admixtures on the durability of rubberized concrete. The results showed that the addition of MK and FA in equal amounts to rubberized concrete yielded the most significant improvement in terms of the ability of rubberized concrete to resist chloride ion penetration (RCPT) and water resistance. On the

basis of the analysis of the microscopic images, it is believed that FA fills the tiny pores caused by the inconsistency between the rubber and the cement matrix during the mixing process of rubber concrete and that MK plays a reinforcing role. The pozzolanic effect of MK activates the cement matrix and strengthens the cement matrix. The hydration reaction in the later stage healed a considerable proportion of the tiny pores. The combination of MK and FA significantly improved the resistance of rubber concrete to chloride ions and water molecules. The effects of three mineral admixtures on the durability of rubber concrete can be seen in Figure 20.

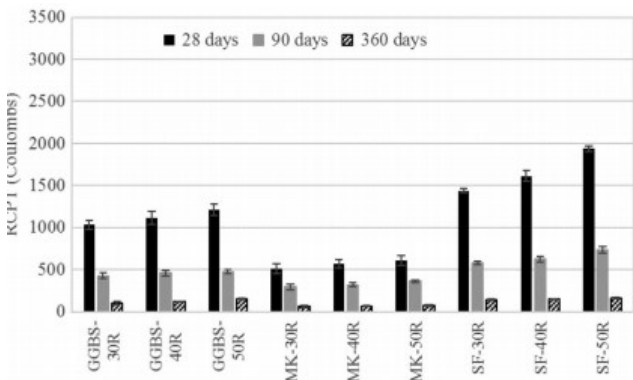

**Figure 20.** Effects of binary mineral admixtures on RCPT results. GGBS-30R FA + GGBS + 30%Rubber content; MK-30R FA + MK + 30%Rubber content; SF-30R SF + FA + 30%Rubber content [17].

The research of Onuaguluchi [16] showed that rubberized concrete mixed with silica fume had better anti-water permeability, and the measured water absorption was smaller. From microscopic research, it is believed that the pozzolanic effect of silica fume in the hydration process of rubber concrete effectively fills the tiny pores inside the specimen, which has been recognized by Basem and Gupta [18,64]. Li [65] mixed carbon nanotubes into rubber cement mortar and studied the changes in compressive strength and water resistance in rubber cement mortar. The test results showed that carbon nanotubes could significantly reduce the water absorption rate of rubber cement mortar, and the absorption rate could be reduced by up to 79% with the optimal dosage. Li measured the water resistance of the specimens for the length of time that the capillary pressure drives the water into the rubber cement mortar specimens. The specimen incorporating carbon nanotubes needed more time, indicating that its water resistance was better than the traditional rubber cement mortar.

Zhang [67] studied the freeze–thaw cycle resistance of metakaolin rubber concrete. The test results showed that metakaolin (MK) could greatly improve the freeze–thaw cycle resistance of rubber concrete, but this improvement was not linear. This characteristic is consistent with the variation trends in compressive strength summarized by the author. The incorporation amount of 13% reached the maximum value. After 100 cycles, the mass loss rate was reduced from 0.60% to 0.34%. The authors believed that the optimal incorporation amount was between 13% and 16%. Of course, after the comparison, it is considered that the improvement in the durability of the concrete by the rubber particles was also significant.

The strong pozzolanic effect of silica fume, metakaolin and other substances helps to improve the efficiency of the hydration reaction inside the concrete specimen, prolong the hydration reaction time and is of great help in improving the performance of rubber concrete. At the same time, the addition of carbon nanotubes and other substances can help rubber concrete build a stronger internal structure and improve its compressive strength and durability at a physical level. The different mineral admixtures selected by different scholars and the actual results are summarized in Table 8.

**Table 8.** Summary table of incorporating external compounds.

| References | Rubber Particle Size | Best Rubber Content | Treatment Methods | Compressive Strength | Durability |
|---|---|---|---|---|---|
| Xue [7] | Maximum size of 6 mm | 5 wt% with silica fume | The silica fume coats the rubber crumb, with free water in the concrete | The compressive strength of rubber concrete decreased with the increase in rubber content, and the compressive strength at 20% replacement rate decreased by 46.68% compared with the control group | Adding rubber to concrete will increase its damping ratio; the addition of silica fume strengthened the adhesion between rubber particles and cement matrix |
| Li [8] | 8–12 mesh | In terms of impact resistance, 30% rubber mixed with 10% micro silicon powder is the best | The silicone rubber modifier is used to pretreat the rubber | The modified effect of the modifier on the coarse-grained rubber is better than that of the fine rubber powder, and the increase ratio of the compressive strength is 20% to 10% | – |
| Hamid [13] | The same size of sand; 10 mm; 20 mm | The combination of ordinary water mixing and 5% rubber replacing coarse aggregate is the best | Adding 7 wt% silica fume into rubber concrete. Rubber concrete prepared by replacing ordinary water with seawater | The compressive strength of the combination of 100% seawater and 20% rubber decreased by 65% compared with the control group | The results of RCP and water absorption test prove that rubber particles can slightly improve the permeability of concrete |
| Mhaya [9] | 1–4 mm, 5–8 mm | 30 wt% mixed rubber | Adding 10–50% ground blast furnace slag as modified material of rubber concrete | The combination of 30% coarse and fine mixed rubber particles had the highest compressive strength after 28 days of curing (27.2 mpa) | The microstructure analysis shows that the increase in rubber content will weaken the interfacial transition zone and cause an increase in porosity |
| Grinys [60] | 0.063–2 mm | For durability, the sample with 10 Kg/m$^3$ rubber replacing fine aggregate performs best | Polypropylene fiber and glass powder mixed into rubber concrete for modification; liquid polymer-based carboxylated styrene butadiene latex used as the surface treatment for rubber particles | The compressive strength of glass powder modified rubber concrete samples after 56 days of curing increased by 13% compared with 28 days, while other samples in the same period increased by only 2.8% | The high specific surface area of fine rubber powder helps to improve the freeze–thaw resistance of rubber concrete. Fine rubber powder can well fill the tiny pores between aggregate and cement matrix |
| Kang [61] | 1–2.36 mm | – | Modification test carried out by adding silica fume into rubber concrete | The addition of rubber reduces the compressive strength of concrete, and the addition of silica fume can help to improve the compressive strength of rubber concrete | The abrasion resistance of rubber concrete is better than that of the control group. The addition of silica fume can improve the abrasion resistance of rubber concrete |

**Table 8.** *Cont.*

| References | Rubber Particle Size | Best Rubber Content | Treatment Methods | Compressive Strength | Durability |
|---|---|---|---|---|---|
| Li [65] | 60 mesh | – | Carbon nanotubes added in the mixing process of rubber concrete | After 28 days of curing, when the CNT content was 0.08%, the compressive strength increased by 57.0%. The compressive strength decreased with the increase in CNT content | CNT effectively improved the impermeability. When the CNT content was 0.04%, 0.08% and 0.12%, the water droplet infiltration rate decreased by 4.3%, 7.6% and 8.7%, respectively, compared with the control group |
| Bashar [66] | 30 mesh | – | Nanosilica added in the mixing process of rubber concrete | The compressive strength of rubber concrete was significantly improved by the addition of nanosilica. The addition of nanosilica reduced the impact resistance of rubber concrete | The addition of nanosilica significantly reduced the porosity of rubber concrete, strengthened the internal microstructure of concrete and improved the durability of rubber concrete |
| Zhang [67] | 40 mesh | The rubber content is 19%, and the partial kaolin content is 13% | Metakaolin was added into rubber concrete as a substitute for cement to study its modification | The addition of rubber reduces the compressive strength of concrete, while the addition of metakaolin enhances the compressive strength of rubber concrete, but there is a threshold for the enhancement effect | After the freeze–thaw cycle, the mass loss of samples mixed with 19% rubber was 0.6% and that of samples mixed with 16% metakaolin was 0.33%, while that of the control group was 4.7% |
| Side [17] | 0.27–0.66 mm, 1.2–2 mm | When the rubber content is 20%, the rubber concrete has the best deflection resistance | Polyvinyl alcohol fiber and polyvinyl alcohol fiber are added to rubber concrete as additives, and fly ash, blast furnace slag, silica fume and metakaolin are used as substitutes for cement | The samples of fly ash, blast furnace slag, silica fume and metakaolin replacing cement all strengthen the compressive strength of rubber concrete | The microstructures of the rubber concrete samples mixed with metakaolin were observed after curing for 360 days. It was found that the pore structure almost disappeared, the water absorption decreased significantly and the durability improved |
| Onuaguluchi [16] | Smaller than 2.3 mm | – | Limestone powder is used as a modified material for rubber. Rubber crumbs, water and LP are mixed in a Hobart mixer at low speed. Coated rubber crumbs are air-dried for 24 h and stored in plastic bags for one month | The compressive strength of rubber and silica fume samples pretreated with limestone powder is much higher than that of ordinary rubber concrete | The addition of rubber reduced the water absorption of concrete, and the samples mixed with silica fume even decreased by 59% compared with the control group |

**Table 8.** *Cont.*

| References | Rubber Particle Size | Best Rubber Content | Treatment Methods | Compressive Strength | Durability |
|---|---|---|---|---|---|
| Basem [64] | Maximum size of 4.75 mm | – | Blast furnace slag, fly ash, silica fume and metakaolin are added into rubber concrete after replacing cement in the proportions of 10%, 20%, 30% and 40% | Compared with the control group, the compressive strength of the samples with 20% rubber content decreased by 58.2%. Silica fume can help improve the compressive strength of rubber concrete | – |
| Gupta [18] | 2–5 mm in width and up to 20 mm in length | – | Fine aggregate is partially replaced by rubber fiber and cement is replaced by silica fume. Use a plasticizer to improve the working performance of concrete | The compressive strength decreases with the increase in the amount of rubber fiber and increases with the increase in silica fume replacement rate | The addition of silica fume will reduce the water absorption of concrete, while the increase in rubber fiber content will increase water absorption. After silica fume replaces cement, the chloride ion permeability of concrete increases |

## 5. Incorporating Fiber Modification

Youssf et al. [6] explored the effects of three different fibers (PP fiber, steel fiber and rubber fiber) on the mechanical properties of rubber concrete. The improvement in compressive strength of rubberized concrete by PP fiber and steel fiber was almost invisible, while the incorporation of rubber fibers had a negative effect (see Figure 21).

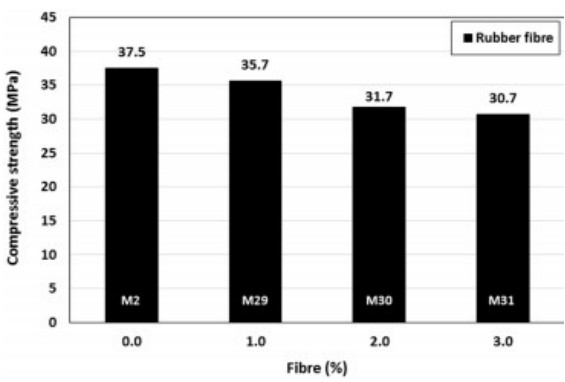

**Figure 21.** Effects of rubber fiber additives on rubberized concrete [6].

Youssf concluded that the rigidity of the rubber fibers was very low. The compressive strength of the rubberized concrete was therefore reduced after their incorporation. FHossain et al. [68] incorporated polypropylene fibers into rubberized concrete and studied the effect of polypropylene fibers on the compressive strength of rubberized concrete. The test results showed that the compressive strength of rubberized concrete was significantly enhanced after adding the fibers, and the maximum compressive strength at 28 days increased by 26.9%. At the same time, Hossain found that the compressive strength of rubberized concrete does not decrease linearly with the increase in rubber content. The combination with 10% rubber content had a better compressive strength performance than the control group composed of ordinary concrete. Then, with the gradual increase in rubber content, the compressive strength of rubberized concrete gradually decreases. This phenomenon

also occurred in Bu's experiments. When examining the experiments in which this kind of strength change phenomenon occurred [69], it was found that it was related to the size of the rubber particles selected in the experiment. Small-sized rubber particles will cause the rubber concrete to deteriorate at a lower dosage. The compressive strength, meanwhile, increased. It is believed that the rubber particles can be distributed in the concrete specimen when the size is small and act as an elastic unit, which can slow down the damage to the internal structure caused by the external load and act as a benign protection unit. However, the increase in the amount of particles incorporated in the rubber will damage the structure to an extent far greater than any protective effect. Carroll et al. [70] explored the effect of polyvinyl alcohol fibers (PVA) on the modification of rubberized concrete. The study showed that the rubberized concrete mixed with PVA had higher compressive strength than the combination without fibers. The results are shown in Figure 22. The authors believe that the incorporation of fibers can help to build a stronger force-bearing system inside rubberized concrete.

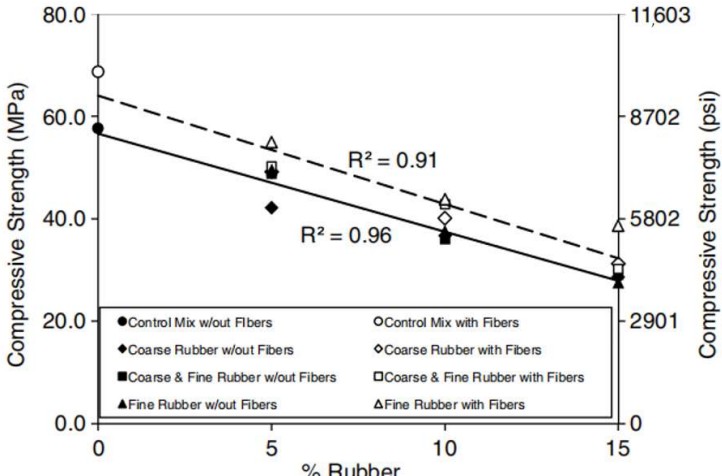

**Figure 22.** Variation diagram of compressive strength of rubber concrete at 28 days with rubber content [70].

Murali et al. [71] proposed a process for preparing rubberized concrete by grouting (see Figure 23). The steel fiber-modified rubber concrete specimens were prepared by grouting method and their compressive strength was studied. The results showed that the steel fiber improved the compressive strength of the rubber concrete, but the effect of the improvement with the low water–binder ratio was far less than that obtained with the high water–binder ratio environment. This was because the steel fibers can better exert their own tie effects in an environment with a high water–binder ratio, and the high water–binder ratio was beneficial with respect to the arrangement of steel fibers in the rubber concrete. The research of Fu [72] also obtained the same conclusion.

Wang et al. [14] studied the modification effect of different kinds of fibers on rubberized concrete. The authors selected four different kinds of fiber materials: short-straight steel fibers (S-1), long hooked-end steel fibers (S-2), long flat-surface synthetic fibers (P-1) and long rippled-surface synthetic fibers (P-2). The results showed that the effect of S-1 fibers was the best; the compressive strength of the 28-day-old specimen could even exceed that of ordinary concrete without rubber, which shows that suitable fiber material can compensate for the mechanical property weakening of concrete caused by rubber.

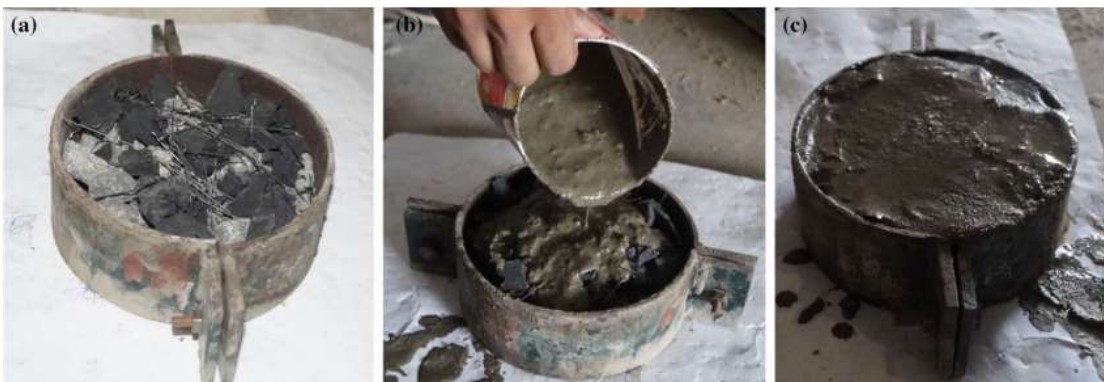

**Figure 23.** Preparation process of rubber concrete. (**a**) Arrange coarse aggregate, rubber and steel fibers. (**b**) Gravity method. (**c**) Sample [71].

Srivastava et al. [73] added carbon fiber into rubberized concrete. The research results showed that 4%, 5% and 6% of carbon fiber rubberized concrete showed good compressive strength performance, and the compressive strength increased with the increase in carbon fiber dosages. Chen et al. [74] studied the combined effect of steel fiber and nanosilica on the mechanical properties of rubber concrete. The authors proposed that 1% is the optimal dosage of steel fiber, while nano-silicon will be further optimized on the basis of strengthening the compressive strength of rubber concrete by steel fibers, and these two things can play an excellent synergistic effect. In the high-temperature environment, the concrete specimen was still affected by the bridging effect of the steel fibers and still could exert a high strength, and the effect of the steel fibers on the high content of rubber concrete was more obvious. However, the enhanced role of nanosilica was gradually lost in the case of high rubber content. Liu [75] studied the feasibility of using three kinds of fiber-modified rubber concrete as bearing components. In the experiment, the author simulated and studied several performance requirements that need to be considered for bearing components. The test results showed that steel fibers, carbon fibers and polymer acrylic fibers can enhance the compressive strength of rubberized concrete.

Wang et al. [14] also focused on the effect of various fiber materials on the surface resistivity of rubberized concrete, and the results showed that the incorporation of fibers had a negative impact on the surface resistivity of rubberized concrete, though the results were still better than those for ordinary silicon concrete. The resistivity values of ordinary concrete were between 6 and 29 k$\Omega$·cm. The rubber concrete modified by fiber still remained between 31 k$\Omega$·cm and 36 k$\Omega$·cm, and the ordinary rubber concrete values were as high as 38 k$\Omega$·cm. Although the incorporation of fibers will have a certain negative effect on the resistivity value of rubberized concrete, this negative effect is acceptable considering that fibers can greatly improve the compressive strength of rubberized concrete. The resistivity value can be considered as the resistance of the concrete interface to chloride ion penetration. A higher resistivity value means better chloride ion penetration resistance. Luo et al. [76] studied the freeze–thaw cycle resistance of steel fiber-modified rubberized concrete. The test results showed that the addition of steel fibers could significantly improve the residual strength of rubberized concrete after freeze–thaw cycles. With the increase in the number of freeze–thaw cycles, the reinforcing effect of steel fibers constantly weakened. At the same time, the authors showed that the mass loss of the rubberized concrete without steel fibers during the freeze–thaw cycle was faster than that of the rubberized concrete with steel fibers.

Turatsinze et al. [77] studied the synergistic effect of rubber and steel fibers in the process of cracking resistance. The test results showed that steel fibers could effectively retain the residual strength of rubber concrete after being damaged by external forces, while rubber could effectively reduce the surface cracks generated in the process of loading. Wang et al. [78] showed that polypropylene fiber could help rubber concrete to build

structures effectively resistant to electron passage. The resistivity of specimens at the age of 28 days is shown in Figure 24. The combination of 10% rubber and 0.5% polypropylene fibers has the best surface resistance and freeze–thaw resistance. The author considers that the alkali–silicon reaction between polypropylene fibers and the hydration process is an important link in improving the durability of rubber concrete.

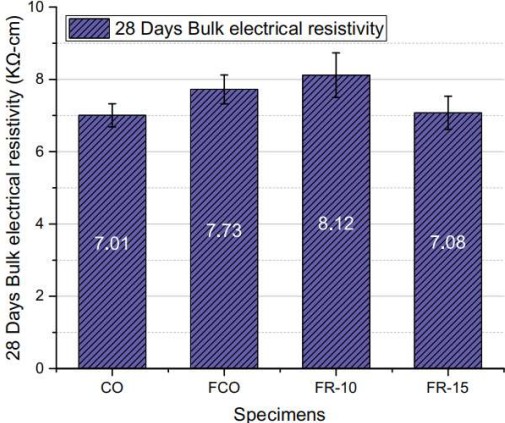

**Figure 24.** The bulk electrical resistivity of concrete specimens [78].

Gupta et al. [15] studied the durability of concrete mixed with rubber fiber and rubber particles, and the results showed that the reinforcement in the mixed rubber concrete was less corroded by chloride ions than that in the concrete prepared with single rubber particles. The author also explored the quality loss of rubber concrete with three different acid solutions, and the results showed that mixed rubber concrete was better than ordinary rubber concrete in terms of acid erosion resistance ability. This was because the characteristics of large size and high curvature of rubber fibers can cooperate with rubber particles to construct a more perfect structural network to resist acid entry and erosion.

To sum up, adding fiber materials, such as carbon fiber, PP fibers and steel fibers, can improve the compressive strength of rubber concrete. From the observation of microstructures, it can be found that mixing with fibers can help rubber concrete bear greater external loads. The fibers can help rubber concrete build a more stable internal mesh structure. Special material fibers such as polypropylene fiber play a more special role, and the alkali–silicon reaction of polypropylene fibers can significantly improve the durability of rubber concrete. The tests conducted by scholars by adding different types and amounts of fibers into rubber concrete are summarized in Table 9.

**Table 9.** Summary table of incorporating fibers.

| References | Rubber Particle Size | Best Rubber Content | Treatment Methods | Compressive Strength | Durability |
|---|---|---|---|---|---|
| Hossain [68] | Maximum CR size of 4.75 mm | The samples with 30% recycled aggregate, 5% rubber particles and 2% pp fibers have the best ductility and bending resistance | In recycled aggregate concrete, rubber particles were added to replace fine aggregate, and PP fiber was added for modification | The addition of rubber will reduce the strength of recycled concrete, while the addition of PP fiber will increase it. The compressive strength of concrete with 10% recycled aggregate replacement was better than that of the 30% replacement and control groups. | – |

Table 9. *Cont.*

| References | Rubber Particle Size | Best Rubber Content | Treatment Methods | Compressive Strength | Durability |
|---|---|---|---|---|---|
| Bu [69] | 1–2 mm | 0.1% pp + 0.9% MS (micro-steel fibers) was the best fiber mix | In the experiment, mixed fibers (PP fibers mixed with micro-steel fibers) and polycarboxylic ether-based superplasticiser were used to modify a mineral admixture | After 0.1% pp fiber +0.9%ms fiber was mixed into concrete, the maximum compressive strength of concrete was 33.94 mpa. Compared with the control group, the compressive strength of concrete samples with 20% rubber content decreased by 44.21% | – |
| Carroll [70] | 9.5 mm, 2.4 mm | – | Five kinds of fibers were selected for modification: 9.5 mm and 13 mm (PVA) fibers, 25 mm twisted steel fibers, 25 mm hooked steel fibers and 13 mm straight steel fibers | The compressive strength of concrete samples will decrease no matter the size of rubber particles, but the samples mixed with fiber always maintain a high value | – |
| Murali [71] | 12–19 mm | – | Soak the rubber particles in 10% volume fraction NaOH solution for 0.5 h, wash them with clean water, then let them stand. At the same time, the steel fibers are used as an additive for modification. The author also took the water–cement ratio as the modification variable | The compressive strength of samples decreased with the increase in rubber content and increased with the increase in steel fiber content. The impact resistance increases with the addition of rubber and steel fibers | – |
| Fu [72] | 1–3 mm | The combination of 10% rubber and 0.75% steel fiber has the maximum compressive strength | Steel fibers with 0.5%, 0.75% and 1% volume fractions were selected as the additives for modification | The compressive strength of the sample is determined by the amount of rubber and steel fibers | – |
| Srivastava [73] | 10–50 mm | 50 mm long rubber fibers and 10% content was the best combination | Cut the rubber tire fiber up to 10 mm. This is used to replace the coarse aggregate in the concrete | The addition of rubber will reduce the compressive strength of concrete, but it shows less strength loss when the length of rubber fibers is 50 mm and the percentage of rubber fiber used is 10% | – |
| Chen [74] | 1–2 mm | The optimum volume ratio of steel fiber modification is 1.0%, and the optimum content of nanosilica modification is 1.0% | The effects of steel fiber volume ratio and nanosilica content on the mechanical properties of rubber concrete were mainly studied | Steel fibers can increase the compressive strength of rubber concrete at high temperatures by 103.93%. The improvement of steel fibers on the compressive strength of rubber concrete at a high temperature is much greater than that at room temperature. Nanosilica can play a reinforcing role | – |

**Table 9.** *Cont.*

| References | Rubber Particle Size | Best Rubber Content | Treatment Methods | Compressive Strength | Durability |
|---|---|---|---|---|---|
| Liu [75] | 40–60 mesh | The best combination is a mixing amount of rubber powder less than 5 kg/m$^3$ and of steel fibers less than 15 kg/m$^3$ | Waste rubber powder, polypropylene fibers, carbon fibers and steel fibers were selected for modification | When the amount of rubber powder is less than 5 kg/m$^3$ and the amount of steel fibers is less than 15 kg/m$^3$, the amount of steel fibers increases and the compressive strength of the modified concrete will be greater than that of normal concrete | – |
| Luo [76] | 0.125 mm | 2% steel fibers was best for rubberized concrete in terms of durability | In this paper, the compressive strength and durability of rubber concrete were studied by adding steel fibers | Rubber will reduce the compressive strength of concrete, while steel fibers play a reinforcing role and the reinforcing role of steel fibers decreases with the increase in the number of freeze–thaw cycles | The influence of steel fibers on the mass loss rate of rubber concrete in freeze–thaw cycles is positive |
| Turatsinze [77] | Smaller than 4 mm | – | The synergistic effect of steel fibers and rubber particles on concrete was studied | The addition of rubber will reduce the compressive strength of concrete but significantly improve its brittleness | The presence of rubber greatly reduces the shrinkage of concrete and the presence of steel fibers enhances this effect |
| Wang [78] | 7–30 mesh | 0.5% polypropylene fibers combined with 10% rubber showed the best resistance to chloride ion penetration | The synergistic effect of polypropylene fiber and rubber particle-modified concrete was studied | After adding rubber into polypropylene fiber concrete, the compressive strength decreases significantly, but the residual stress after fracture increases significantly | The synergistic effect of rubber particles and polypropylene fibers greatly improves the freeze–thaw resistance and chloride penetration resistance of concrete |
| Gupta [15] | Width of 2–5 mm and length up to 20 mm, 0.15–1.9 mm | The combination with 10% rubber particles had the lowest mass loss in the freeze–thaw cycle test | The modification effects of rubber particle aggregates and rubber fibers on concrete were investigated | – | The water resistance of rubber particles and fibers helps concrete better resist the corrosion of acid substances. Rubber particles and rubber fibers provide resistance to chloride ions |

## 6. Coefficient of Thermal Conductivity

Qin et al. [19] also paid attention to the change in thermal conductivity of silicone rubber concrete after special treatment. The test results are shown in Figure 25. Silicone rubber concrete is considered to be an excellent thermal insulation material. Based on the analysis of the porosity and compressive strength of silicone rubber concrete, the author thinks that the improvement in thermal insulation performance with silicone rubber concrete is due to the excellent thermal insulation performance of silicone rubber itself, which is less affected by external interference factors. Liu's [75] study also involved the temperature difference resistance of fiber-modified rubber concrete, and the test results showed that rubber concrete itself had a better ability to resist temperature changes compared with ordinary concrete.

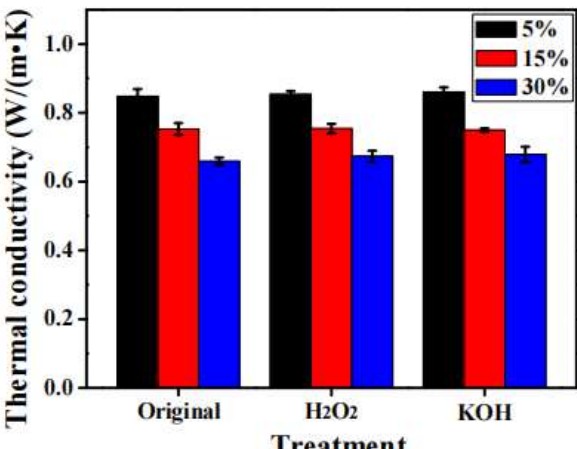

**Figure 25.** Thermal conductivity of original and treated silicone rubber concrete [19].

Marie [20] explored the effect of adding rubber into recycled aggregate concrete on thermal insulation performance. The study showed that the addition of rubber can significantly improve the durability of recycled aggregate concrete, mainly manifested in a reduction in thermal conductivity. The thermal conductivities of the recycled aggregate concrete group and the rubber-incorporated group are shown in Figure 26. It can be observed that with the increase in rubber content and recycled aggregate, both concretes showed better thermal insulation performance, which was due to the thermal insulation performance of recycled aggregate and rubber particles being better than that of ordinary aggregate.

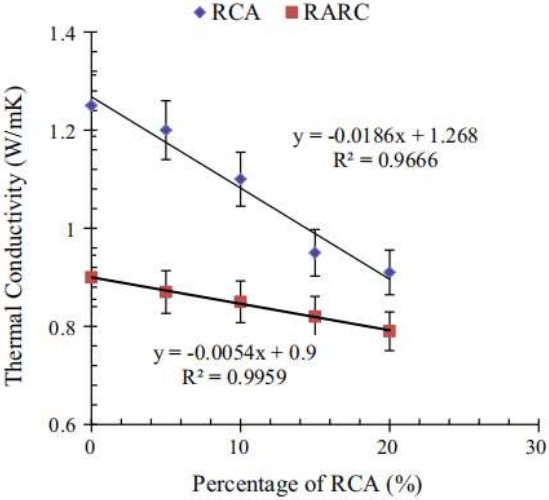

**Figure 26.** Relationship between the thermal conductivity of concrete mixture made with recycled aggregate and rubber aggregate and the replacement rate of recycled aggregate [20].

Petrella et al. [21] studied the influence of rubber particles of different size on the thermal conductivity of recycled aggregate concrete. The results showed that the fine rubber particle concrete had a better thermal insulation ability than the coarse rubber particle concrete. This was because fine rubber particles have a larger specific surface area, which can weaken the fluidity inside the concrete, and the rubber particles do not easily float and segregate, which reduces the weak links in the concrete, which is helpful for the thermal insulation performance of the concrete. Jin et al. [79] compared the aging of rubber concrete mortar and ordinary cement mortar in a high-temperature environment. The research shows that the mortar mixed with rubber and ordinary mortar resisted aging at the same high temperature for the same time, and the former had a better resistance hardness residue. Benazzouk et al. [80] used the self-coherence mean model to discuss the influence of rubber on the thermal properties of concrete. The test results showed that the thermal conductivity of rubber concrete can be reduced by about 80% compared with ordinary concrete, and the minimum can reach 0.47 W/mK. The author believes that rubber has great potential to modify the thermal properties of ordinary concrete.

To sum up, the thermal conductivity of rubberized concrete is lower than that of traditional concrete, which also means that rubberized concrete has better thermal insulation performance. This is due to the excellent thermal insulation ability of the rubber particles themselves. The relevant tests and conclusions on the thermal conductivity and thermal insulation performance of rubber concrete are summarized in Table 10.

**Table 10.** Summary table of thermal conductivity.

| References | Rubber Particle Size | Best Rubber Content | Treatment Methods | Thermal Conductivity |
|---|---|---|---|---|
| Qin [19] | 5, 10, 20 and 50 mesh | 5 wt% treated with KOH or $H_2O_2$ | 1. Soak in $H_2O_2$ for 24 H at room temperature soak in KOH for 24 H, then wash with water several times | Compared with the control group, the thermal conductivity of the sample with 30% silicone rubber content was reduced by 60.64%, and the silicone rubber concrete had good thermal insulation and thermal insulation properties |
| Marie [20] | 0.075–4.75 mm | 20% recycled aggregate instead of coarse aggregate and 10% rubber particles instead of fine aggregate constituted the best combination | The author used recycled aggregate instead of coarse aggregate and rubber powder instead of fine aggregate to explore the modification of concrete | Appropriate recycled aggregate and rubber aggregate can reduce the thermal conductivity by up to 32% compared with the contract control group |
| Petrella [13] | 0.5–2 mm | – | In this paper, rubber particles and waste porous glass were used as aggregate substitutes to modify the properties of concrete | The rubber particles fill the pores between the porous glass and the cement matrix, preventing heat transfer, thus reducing the thermal conductivity of concrete |
| Liu [75] | 40–60 mesh | The best combination was with the mixing amounts of rubber powder less than 5 kg/m$^3$ and steel fibers less than 15 kg/m$^3$ | Waste rubber powder, polypropylene fibers, carbon fibers and steel fibers were selected for modification | The temperature resistance of fiber waste rubber concrete was better than that of ordinary concrete in the control group, which shows that the temperature change in fiber waste rubber concrete was less than that in the control group within the same temperature change range |

**Table 10.** *Cont.*

| References | Rubber Particle Size | Best Rubber Content | Treatment Methods | Thermal Conductivity |
|---|---|---|---|---|
| Benazzouk [80] | Smaller than 1 mm | 50 wt% | – | The thermal conductivity of concrete samples with rubber particles decreases rapidly and decreases with the increase in rubber content |

## 7. Conclusions

This paper summarizes the modification of rubberized concrete with chemical solutions, admixtures, rubber size and fibers, characterized by compressive strength and durability, and also summarizes the influence of concrete on the thermal conductivity of rubber. The paper also summarizes research on waste rubber particles, particle size and properties. The results show that rubber particles pretreated with chemical solutions are more closely bound to the cement matrix, with higher compressive strength and better durability. Compared with the rubber concrete without any treatment, it can be found that the pretreatment can significantly improve the mechanical properties and durability of rubber concrete, and this fact proves that chemical solution pretreatment is effective. The pozzolanic effect of wollastonite and metakaolin in the admixture helps to strengthen the hydration reaction of the rubberized concrete, improve the overall strength of the concrete and reduce the excess pores in the concrete specimen. Additives such as carbon nanotubes can help rubber concrete build a better internal structural system, while fibers can reduce the development of cracks in rubber concrete under load and significantly improve the compressive strength of rubber concrete. The addition of rubber significantly reduces the thermal conductivity of concrete, and rubberized concrete has excellent thermal insulation properties. Rubber concrete can be modified in the above ways, and modified rubber concrete products have the advantage of being suitable for more application scenarios. In addition to the above three modification methods, there are many other aspects to be explored in the research on rubber concrete modification, such as the treatment of rubber particles with high-energy rays and high-temperature heating. The mechanical properties of concrete and rubber obviously decrease after mixing, which limits the range of potential applications. However, on the basis of the current research results for rubber concrete modification, modified rubber concrete has good market application value and is worthy of further exploration.

**Author Contributions:** C.B. was mainly responsible for the grasp of the writing ideas of the article and put forward the innovative points of the article research. D.Z. was in charge of the writing and first translation. L.L. and T.X. were mainly responsible for literature query and download. X.L. and W.Z. mainly responsible for proofreading the translation of articles. Y.S. mainly responsible for the discussion of innovative points of the article and the guidance of writing ideas. L.Y. mainly responsible for answering difficult questions in the process of literature reading, and assisting in writing. All authors have read and agreed to the published version of the manuscript.

**Funding:** This research was funded by a project of the Natural Science Foundation of Chongqing municipality (cstc2021jcyj-msxmX0444) as well as a project of the Chongqing Construction Science and Technology Plan (2021, nos. 1–6) and a project of the Chongqing Bureau of Human Resources and Social Security (cx2020008). This study was also supported by an open fund of the Chongqing Key Laboratory of Energy Engineering Mechanics & Disaster Prevention and Reduction (EEMDPM2021103), a cooperation project of the Ministry of Education "Chunhui Planning" (Z2015147), a postgraduate science and technology innovation project of Chongqing University of Science and Technology (no. YKJCX2120601), and funding from the State Key Laboratory of Bridge Engineering Structural Dynamics, Key Laboratory of Bridge Earthquake Resistance Technology, Ministry of Communications, PRC. Project also supported by scientific and technological research program of Chongqing Municipal Education Commission.

**Institutional Review Board Statement:** Not applicable.

**Informed Consent Statement:** Not applicable.

**Acknowledgments:** Thanks to all those who helped write this review.

**Conflicts of Interest:** The authors declare that they have no known competing financial interest or personal relationships that could appear to influence the work reported in this paper.

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
