# Peer review of "Modification of Rubberized Concrete: A Review"

_buildings, doi:10.3390/buildings12070999_

Round 1

Reviewer 1 Report

This paper focuses on the research status of the modification of rubberized concrete by chemical solutions, admixtures, rubber size, and fibers, characterized by compressive strength and durability, and also summarizes the influence of concrete on the thermal conductivity of rubber.

The title well represents the scope of the review. The abstract well defines the purpose and content of the paper. The structure of the manuscript is coherent and comprehensible.  

I could not find any logical errors, however following points may be considered while revising the article:

1.      Page No 2: Please give the references:

“Some scholars have shown that the excellent water resistance of rubber can enhance the chloride ion penetration resistance and penetration resistance of concrete, and this conclusion has been recognized by many scholars. However, some scholars believe that the micropores generated by the loose bonding between the rubber particles and the cement matrix will weaken the resistance of rubber concrete to the penetration of chloride ions, and the micropores will become the channels for the penetration of chloride ions.”

2.      Please clarify the sentence: “Observing the rubber particles in the aged test block, it is found that different from the initial smooth surface, there is a clear roughness, which indicates that there is a certain substance that corrodes the rubber during the hydration reaction.”

3.      Please check the references. For example: Taak et al.and Mhaya et al. have the same reference (Table2).

4.      Titles of the tables and figures should be without punctuation mark at the end of the sentence, for example: “Table 4. rubber chemical compositions[48].”Also  - in the middle of the text: „Zhong et al. [53]. used styrene-acrylic“ „Chen et al. [75]. studied the combined..“

5.      Should be no punctuation in the middle of the sentence, for example:  „Hamid et al. [56]. studied the mechanical properties of rubber concrete mixed with seawater mixture, and designed 13 different seawater rubber concrete test groups.“

6.      Formatting of the paper should be reviewed.

It was a pleasure to read this manuscript. I wish the author of the best.

Author Response

Comment: 

Page No 2: Please give the references:

“Some scholars have shown that the excellent water resistance of rubber can enhance the chloride ion penetration resistance and penetration resistance of concrete, and this conclusion has been recognized by many scholars. However, some scholars believe that the micropores generated by the loose bonding between the rubber particles and the cement matrix will weaken the resistance of rubber concrete to the penetration of chloride ions, and the micropores will become the channels for the penetration of chloride ions.”

Reply: 

Thank you very much for your suggestion. We have made changes to the places in the article where semicolons should not appear.

Modified:

Some scholars[54-56] have shown that the excellent water resistance of rubber can enhance the chloride ion penetration resistance and penetration resistance of concrete, and this conclusion has been recognized by many scholars[11,73,80]. However, some scholars[60,66-67] believe that the micropores generated by the loose bonding between the rubber particles and the cement matrix will weaken the resistance of rubber concrete to the penetration of chloride ions, and the micropores will become the channels for the penetration of chloride ions.

Comment: 

Please clarify the sentence: “Observing the rubber particles in the aged test block, it is found that different from the initial smooth surface, there is a clear roughness, which indicates that there is a certain substance that corrodes the rubber during the hydration reaction.”

Reply: 

Thank you for your question. We have read the quotation 57 in depth, reviewed the five micro images in Figure 19 repeatedly, and thought that we should not discuss them on the five general diagrams showing the different structures inside the concrete. Finally, we chose and intercepted figure D which is appropriate to some sub topics, and modified the sentences you raised questions.

Comment: 

Please check the references. For example: Taak et al.and Mhaya et al. have the same reference (Table2).

Reply: 

Thank you for your reminding. We have checked all the references in this article in detail and revised them.

Comment: 

Titles of the tables and figures should be without punctuation mark at the end of the sentence, for example: “Table 4. rubber chemical compositions[48].”Also  - in the middle of the text: „Zhong et al. [53]. used styrene-acrylic“ „Chen et al. [75]. studied the combined..“

Reply: 

Thank you very much for pointing out the problem. We have modified the non-standard format in the article..

Comment: 

Should be no punctuation in the middle of the sentence, for example:  „Hamid et al. [56]. studied the mechanical properties of rubber concrete mixed with seawater mixture, and designed 13 different seawater rubber concrete test groups.“

Reply: 

Thank you very much for pointing out the problem. We have modified the non-standard format in the article..

Comment: 

Formatting of the paper should be reviewed.

Reply: 

Thank you for your suggestions. We have further revised the format of the article to meet the next review process.

Reviewer 2 Report

1. Improve Fig.18 (quality)

2. Its not clear to read how much strength is being improved by using pre treatment. Can it become equal to the control sample? 

Indicate which percentage of rubber is suitable for replacement 

3. how do u justify the CNTs as its very expensive

4. conclusion does not refer to the pretreatment as an effective strategy

5. draw more tables if required for clear understanding by the reviewer or reader

Author Response

Comment: 

Improve Fig.18 (quality)

Reply: 

Thank you very much for your suggestions. We have optimized image 18 through better image extraction methods.

Modified:

Figure 18. Comparison among concrete mixtures in terms of 28-day compressive strength (M1 Control/conventional mix with freshwater and natural aggregates; M2 Seawater-mixed naturalaggregate concrete;M3.a Freshwater-mixed; 5% of sand replaced by rubber; M3.b Seawater-mixed; 5% of sand replaced by rubber; M4.a Freshwater-mixed; 10% of sand replaced by rubber; M4.b Seawater-mixed; 10% of sand replaced by rubber; M5.a Freshwater-mixed; 20% of sand replaced by rubber; M5.b Seawater-mixed; 20% of sand replaced by rubber; M6.a Freshwater-mixed; 5% of aggregate replaced by rubber; M6.b Seawater-mixed; 5% of aggregate replaced by rubber; M7.a Freshwater-mixed; 10% of aggregate replaced by rubber; M7.b Seawater-mixed; 10% of aggregate replaced by rubber; M8.a Freshwater-mixed; 20% of aggregate replaced by rubber) [56]

Comment: 

Its not clear to read how much strength is being improved by using pre treatment. Can it become equal to the control sample?

Reply: 

Thank you very much for your question. In fact, even though the strength of pretreated rubber concrete can not be compared with that of the control group without rubber, the strength of pretreated rubber concrete is significantly improved compared with that of the untreated rubber concrete. The degree of improvement is affected by the treatment method, treatment substance and the size of rubber particles. At the same time, the most appropriate rubber content cannot be applied to qualitative analysis, but there should be an optimal value for each quantitative test. We will compare the optimal rubber of all tests mentioned in the article.

Modified:

In fact, even though the strength of pretreated rubber concrete can not be compared with that of the control group without rubber, the strength of pretreated rubber concrete is significantly improved compared with that of the untreated rubber concrete. The degree of improvement is affected by many factors, such as treatment method, treatment substance and rubber particle size.

Comment: 

how do u justify the CNTs as its very expensive

Reply: 

The research on "Study on the Effect of Carbon Nanotubes on Mechanical Properties and Microstructure of Rubber Aggregate Mortar" shows that CNTs are helpful for the connection between aggregates in rubber concrete, mainly in two aspects: 1. Bridging of CNTs 2. The positive effect of CNTs on the hydration reaction. The above two points explain the improvement principle of CNTs on the mechanical properties and durability of rubber concrete.

Comment: 

conclusion does not refer to the pretreatment as an effective strategy

Reply: 

Thank you for your suggestion. In the conclusion, we made clear the effectiveness of chemical solution pretreatment on the performance improvement of rubber concrete.

Modified:

The results show that the rubber particles pretreated with chemical solution are more closely bound to the cement matrix, with higher compressive strength and better durability. Compared with the control group of rubber concrete without any treatment, it can be found that the pretreatment can significantly improve the mechanical properties and durability of rubber concrete, and the fact proves that the chemical solution pretreatment is effective.

Comment: 

draw more tables if required for clear understanding by the reviewer or reader.

Reply: 

Thank you for your suggestions. We have extracted the changes in compressive strength, durability and the best rubber content from the cited literature, and attached them to each paragraph in a table for the convenience of readers.

Round 2

Reviewer 2 Report

The manuscript has been much improved and suitable.